# Phenotypic and Genotypic Virulence Characterisation of *Staphylococcus pettenkoferi* Strains Isolated from Human Bloodstream and Diabetic Foot Infections

**DOI:** 10.3390/ijms232415476

**Published:** 2022-12-07

**Authors:** Chloé Magnan, Nour Ahmad-Mansour, Cassandra Pouget, Madjid Morsli, Sylvaine Huc-Brandt, Alix Pantel, Catherine Dunyach-Remy, Albert Sotto, Virginie Molle, Jean-Philippe Lavigne

**Affiliations:** 1Bacterial Virulence and Chronic Infections, INSERM U1047, Department of Microbiology and Hospital Hygiene, University of Montpellier, CHU Nîmes, CEDEX 09, 30029 Nîmes, France; 2Laboratory of Pathogen Host Interactions, University of Montpellier, CNRS, UMR 5235, 34095 Montpellier, France; 3Bacterial Virulence and Chronic Infections, INSERM U1047, Department of Infectious Diseases, University of Montpellier, CHU Nîmes, CEDEX 09, 30029 Nîmes, France

**Keywords:** *Staphylococcus pettenkoferi*, emerging pathogen, whole-genome sequencing, biofilm study, virulence, genomic analysis, zebrafish infections, resistance

## Abstract

*Staphylococcus pettenkoferi* is a recently described coagulase-negative *Staphylococcus* identified in human diseases, especially in infections of foot ulcers in patients living with diabetes mellitus. To date, its pathogenicity remains underexplored. In this study, whole-genome analysis was performed on a collection of 29 *S. pettenkoferi* clinical strains isolated from bloodstream and diabetic foot infections with regard to their phylogenetic relationships and comprehensive analysis of their resistome and virulome. Their virulence was explored by their ability to form biofilm, their growth kinetics and in an in vivo zebrafish embryo infection model. Our results identified two distinct clades (I and II) and two subclades (I-a and I-b) with notable genomic differences. All strains had a slow bacterial growth. Three profiles of biofilm formation were noted, with 89.7% of isolates able to produce biofilm and harbouring a high content of biofilm-encoding genes. Two virulence profiles were also observed in the zebrafish model irrespective of the strains’ origin or biofilm profile. Therefore, this study brings new insights in *S. pettenkoferi* pathogenicity.

## 1. Introduction

Diabetes mellitus is considered as a global epidemic with almost 550 million adults affected worldwide, a number predicted to rise to 640 million by 2030 [1]. Diabetic foot ulcers (DFU) represent a significant complication of diabetes mellitus, affecting 34% of patients during their lifetime [2]. Initially superficial, these chronic wounds are responsible for frequent infections (more than 50% of cases) [3] spreading to soft tissues and bone structures in 44 to 68% of cases [2]. Bone infection leads to diabetic foot osteomyelitis (DFOM), a real public health problem responsible for one amputation every 20 s worldwide [4]. DFOM and amputations participate in reducing the life quality of patients, increasing the morbidity and mortality, and represent a costly issue for health care systems [2,4,5,6].

The majority of these chronic wounds are polymicrobial [7]. Culturomic and metagenomic approaches revealed that DFU and DFOM ecology is established by a predominance of Firmicutes and particularly *Staphylococus aureus* and coagulase-negative *Staphylococcus* (CoNS) [7,8,9]. While *S. aureus* is clearly implicated in the unfavourable course of these wounds [10], the role of CoNS in infected DFUs remains poorly understood, even though their pathogenicity has been reported [11,12,13,14]. Twelve of the fifty CoNS species, including *Staphylococcus lugdunensis*, *Staphylococcus saprophyticus, Staphylococcus schleiferi* and *Staphylococcus epidermidis*, have been identified as true pathogens and are at the origin of endogenous or nosocomial infections involving bacteremia, pneumonia, urinary tract infection, wound infection and osteomyelitis. These bacteria, constituting the cutaneous commensal microbiota, are considered as opportunists and present different virulence potentials. It is therefore essential to investigate their role in bacterial pathogenicity.

First described in 2002 [15], *Staphylococcus pettenkoferi* is a CoNS increasingly described in human pathology, especially in bloodstream infection [16,17,18,19,20,21,22,23,24] and osteomyelitis [13]. This skin commensal and opportunistic bacterium mainly infects immunocompromised patients or patients with comorbidities such as diabetes mellitus [24,25]. Thus, *S. pettenkoferi* is notably a frequent bacterium isolated from DFOM, suggesting that it could also be involved in the pathophysiology of these infections [26]. Recently, our team described the virulence of one clinical strain isolated from this condition and, notably, its ability to proliferate in phagocytes and to persist inside macrophages and its virulence was confirmed in the zebrafish model of infection [27].

In this study, we performed a genomic comparison and established the biofilm, resistance and virulence potential of a collection of clinical *S. pettenkoferi* strains isolated from both bloodstream and infected DFUs.

## 2. Results

### 2.1. Genetic Diversity of Clinical S. pettenkoferi Strains

We analysed a collection of 29 *S. pettenkoferi* strains isolated from infected DFUs or DFOM (*n* = 22) and blood cultures (*n* = 7) between July 2018 and June 2022 at Nîmes University Hospital (France). *S. pettenkoferi* genome sequences of these clinical strains, the reference strain CIP 107711 and five available *S. pettenkoferi* draft genomes, including the genome reference FDAARGOS_288 (CP022096), were used for genome-based phylogenetic comparison (Figure 1). Whole-genome sequencing (WGS) bioinformatic data are summarised in Appendix A.

Two main clades were determined. Clade I contained two subclades. Six strains from both origins (SP165, NSP008P, NSP009P, NSP019P, NSP003H and NSP005H) clustered together with CIP 107711, 1286_SHAE and 1H7 in subclade I-a. Subclade I-b included the genome reference FDAARGOS_288 and 589_SHAE draft genome and was composed of 15 other clinical strains isolated from DFUs (NSP004P, NSP007P, NSP010P, NSP011P, NSP012P, NSP013P, NSP015P, NSP016P, NSP017P, NSP018P, NSP021P, NSP022P, NSP023P, NSP024P and NSP026P) and four bloodstream isolates (NSP001H, NSP002H, NSP004H and NSP007H). Clade II included the four other clinical strains (NSP008H, NSP003P, NSP020P and NSP025P). Inter-clade diversity was <5% sequence divergence from each other. In contrast, intra-clade diversity was <3%, <2% and <1% for I-a, I-b and II, respectively.

### 2.2. Resistome of S. pettenkoferi Strains

The main characteristics and resistance profiles of the 29 *S. pettenkoferi* clinical isolates are listed in Table 1.

Thirteen resistance-associated genes were reported after genomic analysis of the 29 *S. pettenkoferi* genomes (Appendix A). Resistome comparisons between strains isolated from infected DFUs and blood cultures are summarised in Table 2.

No difference was noted between the resistome content and the results of susceptibility testing. Moreover, this resistome content was not significantly different between strains isolated from blood culture and infected DFUs. Eleven clinical isolates (41.4%) were resistant to penicillin G (with the presence of the *blaZ* gene) and only one strain (SP165) carried the *mecA* gene and was methicillin-resistant. All isolates were susceptible to gentamicin. The only resistance to aminosides was the presence of the *ant(9)-Ia* gene, encoding an aminoglycoside nucleotidyltransferase, and presenting in three strains isolated from infected DFUs. The SP165 strain was resistant to rifampicin with a mutation in the *rpoB* gene (p.H481N). NSP009P was resistant to tetracyclin, with the *tet(K)* gene detected in its genome. The *ermA* gene, conferring resistance to erythromycin and clindamycin, was only detected in 10.3% (*n* = 3) of the strains, while the *vgaA* gene, conferring resistance to streptogramin A and related antibiotics, was identified in 20.7% (*n* = 6) of the strains. Concerning resistance to quinolones, 20.7% (*n* = 6) of the genomes harboured a mutation in *gyrA* (corresponding to p.S84L in gyrase A protein) and 76.7% (*n* = 23) in the *glrA* (corresponding to p.S80Y in topoisomerase IV subunit A protein) gene. Strains isolated from infected DFUs more frequently possessed a *grlB* mutation (corresponding to pE422D in topoisomerase IV subunit B protein) than strains isolated from blood cultures (59% vs. 14.3%, respectively). The *fusB* gene was detected in 26.7% (*n* = 8) of the genomes. Among the 14 strains resistant to fosfomycin, 10 harboured the *fosB* gene, which was found in 37.9% (*n* = 11) of the studied genomes. Five isolates (17.2%) were multidrug resistant (i.e., resistant to ≥3 antimicrobial groups) (SP165, NSP009P, NSP004P, NSP012P and NSP001H).

### 2.3. Biofilm Capacity of S. pettenkoferi Strains

#### 2.3.1. Ability to Form Early Biofilm

The ability of 29 *S. pettenkoferi* clinical strains to form biofilms was analysed by the Biofilm Ring Test^®^ (Biofilm Control, Saint-Beauzire, France) and compared against the CIP 107711 reference strain. Biofilm formation index values (BFI) evaluating *S. pettenkoferi* adhesion ability were distributed in three distinct groups: BFI values = 0 at 4 h of incubation for Group 1; BFI values < 5 at 5 h of incubation for Group 2; and BFI values ≥ 5 at 5 h of incubation for Group 3 (Figure 2). Interestingly, the majority of *S. pettenkoferi* strains (*n* = 17, 58.6%) were able to produce an early biofilm after 4 h of incubation in BHI and were thus classed to Group 1. Group 2 contained nine (31.0%) strains isolated from infected DFU which required a longer incubation time to reach BFI values near 2 at 5 h (a value considered as fixed biofilm in this assay). Finally, three (10.3%) strains constituting Group 3 presented low adherence and potential of biofilm formation. The reference strain CIP 107711 represented our negative for biofilm production with BFI values around 20 during the whole experiment. The three groups showed significant differences in the change in BFI values over time (Mann-Whitney test; *p* < 0.001). Interestingly, all strains isolated from blood cultures belonged to Group 1, with strong production of early biofilm.

#### 2.3.2. Biofilm-Related Genes within *S. pettenkoferi* Genomes

Biofilm was further studied by determining the biofilm-encoding genes in all 29 *S. pettenkoferi* genomes. The coding sequences (CDS) annotated were assigned functions based on Best DIAMOND Hit using the UniProt Reference Clusters (UniRef50) database. The prevalence of these biofilm-associated genes in the 29 studied *S. pettenkoferi* genomes according to the biofilm production of the strains and their origin (blood cultures or DFUs) are presented in Table 3.

All strains harboured the *atlE* gene, encoding for autolysin E, a major contributor to primary attachment to abiotic surfaces; the *BUY34_0850* gene encoding for phenol soluble modulin beta protein (known for its major role in biofilm detachment); the *splF* and *htrA* genes encoding for serine proteases involved in biofilm structuration; and the *clpP* gene encoding for ClpP protease, implicated in *S. epidermidis* biofilm development. The poly-N-acetylglucosamine (PNAG) production is dependent on the presence of the intracellular adhesion (*ica*) operon for which the *icaA*, *icaB* and *icaC* genes were identified in all strains, while *icaD* and *icaR* were absent. We also noted a high presence of the most important biofilm regulators genes (*agrA* and *agrC*, being part of the Agr quorum sensing system; the *rsbUVW-sigB* operon; *mgrA*; *sarA*; *luxS*; *lytSR*; *alrRS*; *cidA*; and *lgrA*).

Interestingly, no significant differences were observed between the presence of biofilm-encoding genes and the profile of the biofilm producers, nor between the origin of the strains (DFU and blood culture) and the clades. Finally, a previous study on *S. saprophyticus* strain MS1146 described the presence of the *uafB* gene, encoding for an adhesin involved in biofilm, on the pSSAP1 plasmid [28] (Appendix A). Notably, this plasmid and the *uafB* gene were present in strains belonging to the two biofilm-producing groups (9/17, 52.9% in Group 1 and 4/9, 44.4% in Group 2) but absent in bacteria belonging to the Group 3.

### 2.4. Fitness of S. pettenkoferi Strains

Based on our biofilm results, we selected three strains isolated from infected DFUs representing the three groups identified by the BRT^®^ experiments: NSP003P (clade II, biofilm Group 1, NSP009P (clade I-a, biofilm Group 2) and NSP0023P (clade I-b, biofilm Group 3), as well as one strain isolated from blood cultures, NSP004H (clade I-b, biofilm Group 1). *S. aureus* reference strain Newman [29] was used as control. Growth profiles of these strains were monitored for over 24 h (Figure 3).

The growth of *S. aureus* Newman was significantly faster than the *S. pettenkoferi* strains as illustrated by the maximum absorbances measures in the stationary phase (Ym) (Ym, *S. aureus* Newman = 1.644, Ym, NSP003P = 1.234, Ym NSP009P = 1.062, Ym NSP023P = 1.086 and Ym, NSP004H = 0.953 respectively, disjoint interval confidence; Appendix A). The comparison between each *S. pettenkoferi* strain showed that NSP0023P presented a significantly higher Ym whereas NSH004H had the lowest. No significant differences between strains were observed for the lag times (K) and inflexion points (1/K) excepted for NSP023P and *S. aureus* Newman (K NSP023P = 0.246 and K *S. aureus* Newman = 0.194; 1/K NSP023P = 4.066 and 1/K *S. aureus* Newman = 5.147, disjoint interval confidence). 

Moreover, a significant difference in slopes was observed between all the *S. pettenkoferi* strains compared to *S. aureus* Newman (Wilcoxon test; *p* = 0.009), confirming that bacterial growth was slower for *S. pettenkoferi* strains. Different growth rates were also observed between NSP004H and NSP003P with a significant slope observed for NSP004H, indicative of a lower growth rate (Dunn test; *p* = 0.03).

### 2.5. Virulence Study of S. pettenkoferi Strains Using a Zebrafish Embryos Model

In vivo virulence of the four selected *S. pettenkoferi* strains (NSP003P, NSP009P, NSP023P and NSP004H) with different biofilm productions and origins was tested using the bath infection model of wounded zebrafish embryos, as previously described for *S. pettenkoferi* [27]. The viability of zebrafish embryos was monitored for 48 h and we observed that mortalities in injured larvae infected with *S. pettenkoferi* began at 18 hpi for all strains (Figure 4). Strains showed different virulence profiles: deaths increased up to 95% by 24 hpi for the NSP009P strain, while embryos infected by NSP003P, NSP0023P and NSP004H showed a mortality of around 50% at 36 h (Log-rank Mantel–Cox statistical test, *p* < 0.001). *S. pettenkoferi* NSP009P was two times more virulent than the other three strains with less than 5% embryo survival at 24 hpi. Interestingly, NSP009P displayed the same virulence profile as for the *S. pettenkoferi* SP165 isolate previously described [27].

### 2.6. Genomic Analysis of Putative Virulence Factors of the S. pettenkoferi Strains

Due to the evidence that the *S. pettenkoferi* strains are virulent and presented two profiles, we investigated their virulome content. A total of 44 putative homologues were identified by genomics analyses (Table 4). Most detected genes were related to immune system evasion and invasiveness (e.g., *capA, eno, katA, tuf, esXA* and five genes encoding for the T7SSb apparatus), whereas most of the toxins-encoding genes were absent (e.g., *hly, hlb, hlg, sea, sel, seg, tst-1, etA, etB, etD* and *lukDE*).

However, three CDSs annotated as haemolysin H1U, delta haemolysin and haemolysin III were identified in 15 (51.7%), 23 (79.3%) and 29 (100%) of all the genomes, respectively. Haemolysin H1U homologue was present in almost all bloodstream isolates (*n* = 6/7) and was only present in biofilm-producing strains.

Additional predicted virulence factors identified in all genomes were associated with lipoprotein maturation (*lspA* and *lgt*), proteases (*clp* proteases), iron scavenging (*isdF*) and nuclease activity (*nucH*). Moreover, two-component systems (TCS) involved in virulence regulation (*saeSR* and *vraSR*) were identified. Conserved virulence factor B and C were present in all genomes, and two isolates (NSP008P and NSP0023P) harboured, in addition, the *virE* gene.

Overall, no significant differences were observed in the virulome content, in the biofilm profiles, or related to the origin of the studied strains. Moreover, regarding the NSP009P strain, which presented a highly virulent profile in the zebrafish model, we could not identify any difference in its virulome content in comparison to the less virulent NSP004H, NSP003P and NSP023P strains. Interestingly, using an alignment of the NSP004H, NSP003P, NSP023P and SP165 genomes against the NSP009P genome, we could characterize conserved genomic regions between the two more virulent strains (SP165 and NSP009P) and the others (Figure 5). Seven regions were identified that will require further investigations to identify putative virulence trends that could explain their different virulence profiles. Genomic diversity was investigated by pangenome and SNP analysis (Appendix A). These analyses confirmed the genomic diversity of the five studied *S. pettenkoferi* isolates (Appendix A). The two most virulent strains (NSP009P and SP165) were clustered together (Appendix A). A similar clustering was also observed with the two less virulent strains NSP004H and NSP023P. Comparison between phenotype (biofilm and zebrafish) and genotype results targeting genes encoding biofilm (Appendix A) and virulence factors (Appendix A) did not identify any specific virulence traits in the two most virulent strains These strains shared conserved coding regions which will require further investigation (Appendix A).

## 3. Discussion

Chronic non-healing wounds are a major complication of diabetes mellitus, associated with severe outcomes such as high morbidity and health care costs [4,30]. Polymicrobial colonisation, biofilm formation, and collaboration between pathogenic and colonising bacteria are hypothesised to impair healing of DFUs, thus contributing to severe complications such as osteomyelitis and amputation [2,3].

The microbial community present on foot ulcers in patients living with diabetes is highly complex and diverse, with CoNS frequently isolated from DFOM specimens [8,9]. Whereas CoNS have long been considered as colonising bacteria belonging to cutaneous commensal microbiota, some species such as *S. lugdunensis* have been identified as true pathogens [14]. However, the role of each CoNS species in infected DFUs remains unclear to date [14,31]. Among them, *S. pettenkoferi* was reported to cause severe infections [13,16,17,18,19,20,21,22,23,24], predominantly in immunocompromised patients. Recently, we determined the pathogenicity of one *S. pettenkoferi* clinical isolate from DFOM, and demonstrated its in vivo virulence associated with a large production of biofilm [27]. Here, we confirm that *S. pettenkoferi* is a pathogen by studying a collection of 29 *S. pettenkoferi* clinical isolates from blood cultures and infected DFU.

Biofilm formation is an important pathophysiological hallmark in DFU, involving most of the bacteria present in chronic wounds, and inducing prolonged inflammation and chronicity of the lesions, as well as the development of antibiotic resistance resulting in difficulties treating these wounds [32]. It has been established that CoNS participate in biofilm formation that could contribute to the chronicity of DFU. However, biofilm-related studies tend to only focus on *S. epidermidis*, as this species is the most frequently isolated [33]. Few studies have evaluated the biofilm formation of *S. pettenkoferi* and, surprisingly, the previous investigations on clinical, animal or environmental isolates reported no production of biofilm [34,35]. We corroborated that the reference strain CIP 107711 was unable to form biofilm as previously published [35]. However, our results showed that clinical *S. pettenkoferi* strains were able to form biofilm, as previously published for one isolate, SP165 [27]. We could classify the strains in three different groups with distinct biofilm behaviours: a large majority of strains (*n* = 17 and the SP165 isolate) corresponding to fast and high producers, a second group of strains (*n* = 9) with an intermediate profile, and three strains with a low potential for biofilm formation. This classification confirmed that most clinical *S. pettenkoferi* strains (*n* = 26, 89.7%) were able to form biofilm. Notably, all strains isolated from blood cultures belonged to the faster biofilm producers group, which is consistent with previous studies showing that 52–100% of CoNS species isolated from bloodstream formed a biofilm [36,37,38].

To corroborate our results, *S. pettenkoferi* genome analysis identified genes known to participate in biofilm formation. All *S. pettenkoferi* genomes harboured the *atlE* gene encoding for the homologous autolytic enzyme AtlE, described in *S. epidermidis* [39]. This protein is a major contributor to primary attachment to abiotic surfaces and a critical enzyme required for the release of eDNA, a component of the staphylococcal in vitro biofilm matrix [40]. Moreover, the NSP012P and NSP023P strains possessed the gene encoding for the Bap homologue protein (Bhp), described in *S. epidermidis* [41]. In *S. aureus*, this protein has been identified as the main determinant for successful surface adherence, as well as intercellular accumulation during biofilm development [42,43]. In addition, the PNAG, a polysaccharide synthesized by products of the *icaADBC* locus, is a well-known mechanism of intercellular adhesion in staphylococcal biofilms. Some genes of this locus (the *icaA, icaB* and *icaC* homologue genes) were identified in all studied strains, although the *icaD* and *icaR* genes were absent. Interestingly, a similar analysis was reported by Dutta et al. (2018) in *S. pettenkoferi* isolated from a cat peritonitis case [34]. The *icaADBC* locus organisation may differ between staphylococci, as *S. lugdunensis* also does not contain the *icaR* gene [44]. Moreover, the *ica* operon expression can be activated by *sarA* and partially by the *sigmaB* and *srrAB* two-component systems [45], and such homologues of major staphylococcal biofilm-regulator-encoding genes were detected in the *S. pettenkoferi* genomes. This included positive regulators such as sigmaB (*sigB*), expressed from and regulated by the *rsbUVW-sigB* operon also present in *S. pettenkoferi* genomes; *sarA*, critical for biofilm formation mainly through repression of proteases; and *mgrA* and *luxS,* two negative regulators. However, the major regulatory system of staphylococcal biofilm formation is the accessory gene regulator (Agr), known to regulate the production of proteases and phenol-soluble modulins (PSMs), main contributors to biofilm maturation and disassembly in *S. aureus* and *S. epidermidis* [46,47]. Phenol-soluble modulin beta (PSMβ) protein and ClpP protease homologues were also annotated in all *S. pettenkoferi* genomes. PSMβ is a key effector in biofilm structuring and dispersion, both in vitro and in vivo [48]. A role in biofilm development has also been identified for the *S. epidermidis* protease ClpP and its expression is regulated by the *agr* system [49]. Finally, the pSSAP1 plasmid described in the *S. saprophyticus* strain MS1146 and containing the adhesin-encoding gene *uafB* was also detected in all the strains. The UafB adhesin is known to contribute to cell surface hydrophobicity and to mediate adhesion to fibronectin and fibrinogen [28]. Therefore, this adhesin could play a role in biofilm formation in *S. pettenkoferi* strains, as it is present in 50% of the biofilm producers’ genomes and absent in the non-producer group.

In accordance with Ahmad-Mansour et al. [27], we confirmed that *S. pettenkoferi* strains showed relatively slow growth compared to *S. aureus*. One hypothesis predicts a positive association between the evolution of host immunity and pathogen virulence involving an increase in replication rates in response to host resistance [50]. Remarkably, a specific set of putative virulence factors (e.g., *capA, katA* and *nucH*) involved in immune evasion were identified in all *S. pettenkoferi* strains. These immune evasion determinants and the slow growth replication of *S. pettenkoferi* may be involved in its persistence during host infection.

Our study confirmed that *S. pettenkoferi* is a virulent pathogen in the zebrafish model of infection, which represents an important animal model to study host–pathogen interactions in vivo [51]. Zebrafish embryos possess a fully functional innate immune system and have been used for their ability to closely mimic human diseases [52]. Bacterial infections in zebrafish embryos are most frequently performed by microinjection. As *S. pettenkoferi* was isolated from chronic wounds, we used an adapted model where a bath infection model was chosen with larvae fin excision to mimic “sterile” wounding damage and inflammation [53]. Immersion following the larvae injury model was recently validated to study *S. pettenkoferi* virulence [27]. The studied strains with different origins and biofilm production ability showed virulence in this model. However, two clear virulence profiles were observed with one strain (NSP009P) being highly virulent, as previously published for the SP165 isolate [27], while the other three studied strains displayed a less virulent profile. However, no difference was observed in virulome content nor in biofilm behaviour. In addition, our genome phylogenetic analysis obtained by WGS data confirmed the presence of distinct clusters and subclusters, each comprising closely related *S. pettenkoferi* isolates from the two different clinical origins. A previous phylogenetic study between 25 *S. pettenkoferi* strains showed that clinical isolates were closely related to environmental isolates, irrespective of their geographic origin [35]. Interestingly, despite the single geographical origin of the strains analysed in this study, a similar repartition between their strains and our strains was observed except for clade II, absent in this previous study. This clade corresponds to a new group of *S. pettenkoferi*, genetically distant from the other strains, and will require phylogenetic investigations on a larger scale. More importantly, no links could be highlighted between biofilm production groups nor with strain origin and cluster membership. However, the two highly virulent strains SP165 and NSP009P are closely related in the same clade, I-a, with seven conserved regions, though more investigation is needed to determine if these regions could correspond to pathogenicity islands.

## 4. Materials and Methods

### 4.1. Bacterial Strains

All *S. pettenkoferi* strains belonged to the collection of the Department of Microbiology at Nîmes University Hospital (France). Twenty-two *S. pettenkoferi* strains were isolated from infected DFUs or DFOM and seven from blood cultures between July 2018 and June 2022. Bacterial identification was obtained by mass spectrometry (Vitek-MS^®^, Biomérieux, Marcy-l’Étoile, France). Antimicrobial susceptibility testing was performed by the disc diffusion test according to the European Committee for Antimicrobial Susceptibility Testing (EUCAST) recommendations [54]. Vancomycin, teicoplanin and daptomycin MICs were determined using broth microdilution procedures (UMIC) (Bruker, Champs-sur-Marne, France). The *S. pettenkoferi* reference strain CIP 107711 (Culture Collection University of Gothenburg, Department of Clinical Bacteriology, University of Gothenburg, Sweden) was used as control in biofilm assays. The *S. aureus* reference strain Newman [29] was used as control in growth curves tests.

### 4.2. Whole-Genome Sequencing and Bioinformatics Analysis

*S. pettenkoferi* strains were cultivated aerobically at 37 °C for 48 h on Columbia sheep blood agar plates (5%) (Biomérieux). Following the manufacturer’s instructions, genomic DNA was extracted using the DNeasy UltraClean Microbial Kit (Qiagen, Aarhus, Denmark). Whole-genome sequencing (WGS) was carried out using Illumina MiSeq sequencing equipment (Illumina, San Diego, CA, USA) using paired-end (PE) read libraries (PE250) made with an Illumina DNA Library Prep Kit (Illumina) according to the manufacturer’s instructions. To examine data quality, raw readings were processed using FastQC (v.0.11.9). Sequencing reads were quality-trimmed and assembled de novo using CLC Genomics Workbench 22.0.2 (Qiagen) with default parameters to include only contigs > 1000 nt. Each *S. pettenkoferi* contig sequence was then annotated with Find Prokaryotic Genes, Annotate CDS with Best DIAMOND Hit and Annotate CDS with Pfam Domains functions within the Functional Analysis tool of the Microbial Genomics Module on the CLC Genomics Workbench with settings as previously described [55]. Virulence and biofilm genes were identified from each annotated *S. pettenkoferi* genome. ResFinder (2022-06-22), PointFinder for *S. aureus* (2019-08), CARD and the NCBI Bacterial Antimicrobial Resistance Reference Gene (2021-08) databases were also used to identify antimicrobial resistance genes [56,57,58]. Sequences obtained were aligned by using the multisequence alignment tool in CLC Genomics Workbench 22.0 with default parameters. Phylogenetic relationships were inferred from this alignment using a distance matrix generated under the assumptions of the Jukes and Cantor model [59]. The phylogenetic tree was constructed by using the neighbour-joining method implemented in the software [60]. WGS was subjected to other investigations, such as circular genome representation using the Proksee programme with CGView.js as its genome drawing engine [61]. Whole-genome bioinformatic data are summarised in Appendix A. All the identified genomics sequences have been deposited on the GenBank website accession bioproject: PRJNA870410.

### 4.3. Kinetics of Early Biofilm Formation

To evaluate the capacity of *S. pettenkoferi* to form biofilm, we used the Biofilm Ring Test^®^ technique (Biofilm Control, Saint-Beauzire, France) following the manufacturer’s recommendations [62]. This assay is based on the measurement of the mobility of superparamagnetic microbeads embedded in biofilms subjected to a magnetic field. Standardised bacterial cultures were deposited on 96-well microtiter plates (Falcon 96 Flat Bottom Transparent, Coming, USA) containing magnetic beads (TON004). The plates were incubated without shaking at 37 °C. After incubation, 100 µL of liquid contrast (LIC001) solution was added on the top of each well, and plates were inserted for 1 min onto a magnetic block and then into a reader (an Epson Scanner modified for microplate reading). Images of each well were acquired at 3 h, 4 h and 5 h. Data were analysed using the BFC Elements 3.0 software (Biofilm Control), giving a result in the form of a Biofilm Formation Index (BFI). Each experiment was performed in triplicate. A negative control was systematically included in each experiment corresponding to Brain Heart Infusion (BHI) medium and beads without bacterial suspension. BFI values were calculated for each well, ranging from 0 (total bead immobilisation, i.e., strongly adherent cells/strong biofilm formation) to 20 (no bead aggregation, i.e., non-adherent cells/no biofilm formation in the experiment conditions). A high BFI value (>7) corresponds to high mobility of the beads under magnetic action (no biofilm), while a low value (<2) corresponds to full immobilisation of the beads due to the sessile cells. Three experiments with two repeats were performed per strain and per incubation time.

### 4.4. Growth Curves Evaluation

Growth profiles were performed in Trypticase Soy Broth (TSB). Then, bacterial suspensions were calibrated to obtain an optical density of around 0.1 at 600 nm. Each bacterial suspension was added to a 96-well plate incubated at 37 °C for 48 h. The absorbance at 600 nm of each well was determined by the Infinite M Mano automatic absorbance reader (Tecan, Männedorf, Switzerland). Each experiment was performed in triplicate. The nonlinear regression model of Gompertz [63] was used with GraphPadPrism 9.2 software (San Diego, CA, USA) to acquire the equations of each growth curve, as well as various notable points (Ym, Y0, K and 1/K). The Gompertz equation can be written as follows:$$Y\left(t\right)=\mathrm{Ym}*\left({\left(\frac{Y0}{\mathrm{Ym}}\right)}^{\exp\left(-K*t\right)}\right)$$
where Y(t) corresponds to the absorbance at a time t, Ym corresponds to the maximum absorbance (stationary phase), Y0 corresponds to the absorbance at t = 0 h, K determines the lag time (1/h), and 1/K (h) is the inflection point of the exponential phase. Inflection points enabled us to determine the beginning of the exponential phase and to calculate the slope of the linear trend line of this phase.

### 4.5. Infection of Danio Rerio Embryos

Following the previous protocol [27], *S. pettenkoferi* strains were cultured in TSB overnight at 37 °C. The cultures were diluted 1:20 and allowed to grow in TSB medium until they reached the mid-exponential growth phase (OD_600_ = 0.7–0.9). Then, the bacteria were centrifuged for 10 min at 4000 rpm, and the pellets were resuspended in fish water with 60 µg/mL of sea salt (Instant Ocean) in distilled water with 4 × 10^−4^ N sodium hydroxide at a concentration of 8 × 10^7^ bacteria/mL. After diluting the inoculum in phosphate-buffered saline (PBS), the samples were plated onto TSB agar plates to determine the number of bacteria present. Experiments were conducted with the GAB zebrafish line in fish water at a temperature of 28 °C. The wounded embryos were placed on a Petri plate, given 0.28 mg/mL tricaine as anaesthesia, and a small tail fin injury was caused prior to infection using a 26-gauge needle under a stereomicroscope (Motic). After tail transection, groups of 24 damaged embryos were placed in Petri dishes with the appropriate bacterial solution (or fish water as a control), and then they were distributed individually into 96-well Falcon plates containing 200 µL of the bacterial suspension (or fish water). Throughout the incubation procedure, the plates were maintained at a constant temperature of 28 °C (bacteria were kept throughout the experiment in fish water, which does not support *S. pettenkoferi* growth). The number of dead embryos could be counted visually based on the absence of a heartbeat.

### 4.6. Ethical Statement

The animal experiments were carried out at the University of Montpellier in accordance with European Union recommendations for the care and use of laboratory animals (https://ec.europa.eu/environment/chemicals/lab_animals/index_en.htm (accessed on 15 March 2021)) and were authorised by the Direction Sanitaire et Vétérinaire de l’Hérault and Comité d’Ethique pour l’Expérimentation Animale under reference CEEALR-B4-172-37 and APAFIS#5737-2016061511212601 v3. Adult zebrafish were not killed for this study, and breeding of adult fish followed the international norms set out by the EU Animal Protection Directive 2010/63/EU. According to the EU Animal Protection Directive 2010/63/EU, all studies were conducted prior to the embryos’ free feeding stage and did not constitute animal experimentation. The cardiac rhythm was used as a clinical criterion for survival graphs. At the end of the survival monitoring, the plates were parafilmed and quickly frozen, stored at −20 °C for 48 h to ensure embryos’ death, and then autoclaved with the bacterially contaminated waste.

### 4.7. Statistical Analysis

Statistical analyses were performed using GraphPad Prism version 9.2 and R software version 4.1.0. Tests used for the *p*-value determination are mentioned in each table and figure legend. A *p*-value < 0.05 was considered to reflect a statistically significant difference.

## 5. Conclusions

This study highlights that *S. pettenkoferi* possesses an important virulence potential as demonstrated in vivo with the zebrafish model of infection, is able to form biofilm and contains a large panel of virulence determinants. Therefore, the genomic analysis of *S. pettenkoferi* constitutes a valuable resource to study molecular mechanisms involved in its pathogenicity.

## Figures and Tables

**Figure 1 ijms-23-15476-f001:**
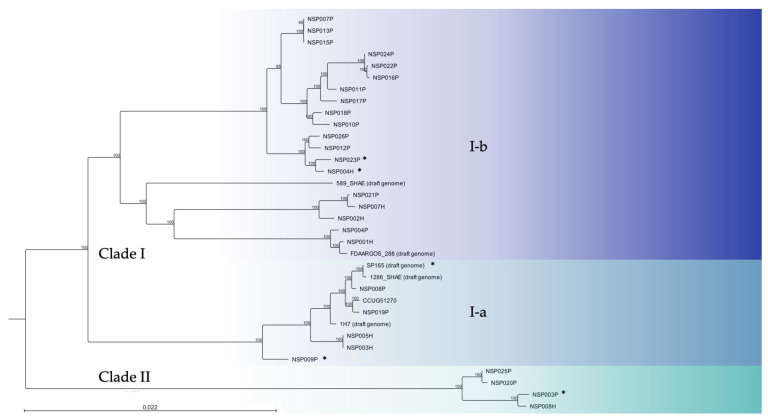
Whole-genome phylogenetic tree of 29 *S. pettenkoferi* clinical isolates aligned with five available *S. pettenkoferi* draft genomes and the CIP 107711 reference strain. Neighbour-joining tree was generated using CLC Genomics Workbench 22.0.2 (QIAGEN Bioinformatics). Numbers at nodes indicate the percentage of bootstrap based on analysis of 100 resampled datasets. The scale bar represents 2.2% sequence divergence. Strains are coloured according to their phylogenetic group (clade). Clade I includes subclade I-a (light blue) and I-b (dark blue). Clade II, green. *, denotes the isolates investigated in depth in this paper.

**Figure 2 ijms-23-15476-f002:**
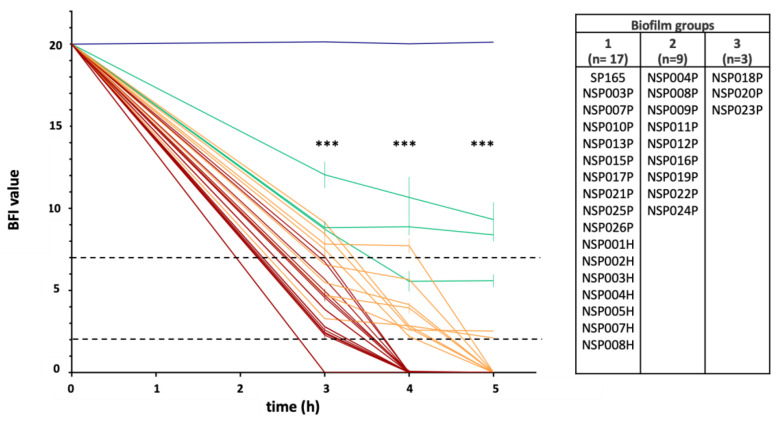
Ability of 29 *S. pettenkoferi* strains to form early biofilm in BHI using BRT^®^ (Biofilm Control)**.** K-means clustering led to three groups: in red, strains with early low biofilm index (BFI) values; in green, low adherent strains; in orange, strains with intermediate BFI values; and in blue, the *S. pettenkoferi* CIP 107711 reference strain with no biofilm production. Dotted horizontal lines: >7, no biofilm; <2, fixed biofilm. Means ± standard errors of the mean of BFIs for at least three independent replicates per group are presented. Statistical differences between BFI values of the three groups at 3, 4 and 5 h of incubation were obtained using Mann-Whitney tests. ***, *p* < 0.001.

**Figure 3 ijms-23-15476-f003:**
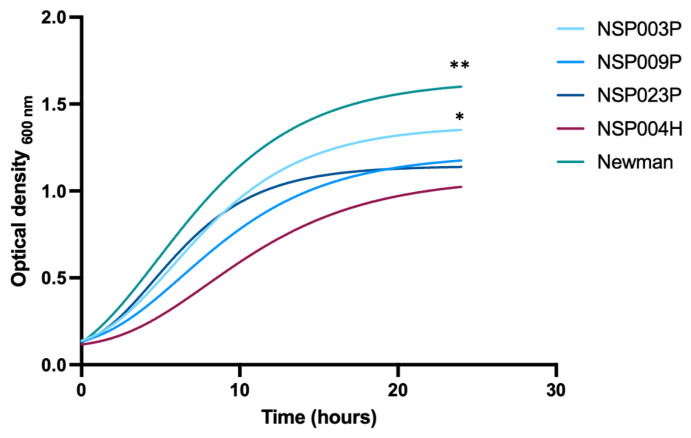
Growth curves of four *S. pettenkoferi* strains compared to *S. aureus* Newman strain using the Gompertz equation. A Wilcoxon test was used to compare the slopes of all *S. pettenkoferi* strains vs. *S. aureus* Newman; significant difference was at ** *p* < 0.01. A Dunn test was used to compare growth rates between the *S. pettenkoferi* strains; significant difference was observed between NSP004H and NSP003P at * *p* < 0.05.

**Figure 4 ijms-23-15476-f004:**
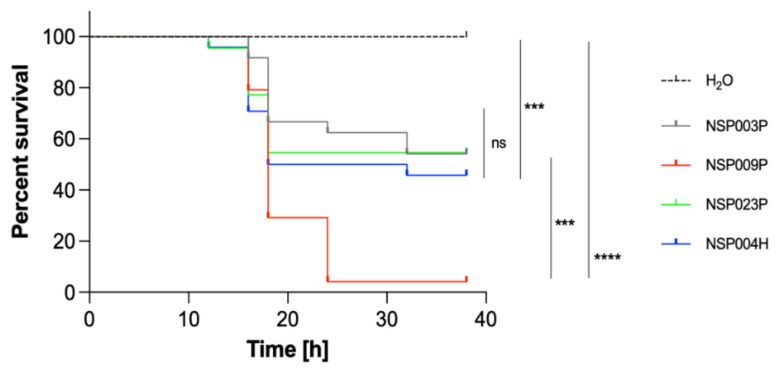
Virulence of *S. pettenkoferi* strains in zebrafish embryos. Kaplan–Meier representation of the survival of zebrafish embryos injured in the tail fin at 48 h post-fertilization (hpf) in a bath infected with *S. pettenkoferi* strains at 8 × 10^7^ CFU/mL grown in exponential phase or “fish water” (negative control). Results are presented as the proportion of surviving embryos (*n* = 24 for each, indicative of four separate experiments). Significant difference at *** *p* < 0.001, **** *p* < 0.0001 or no significant difference (ns).

**Figure 5 ijms-23-15476-f005:**
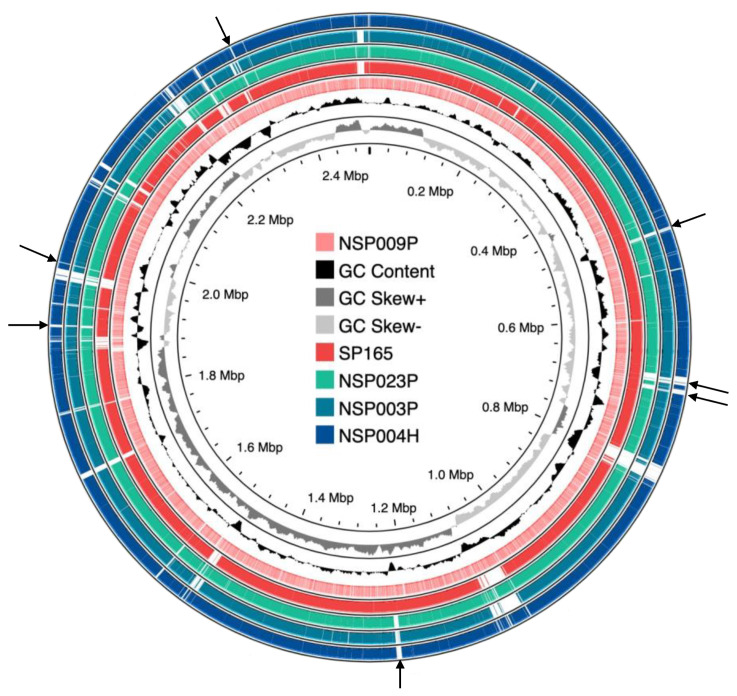
Sequences comparison of *S. pettenkoferi* genomes. The NSP009P strain was compared against SP165 (a virulent strain) and the NSP003P, NSP023P and NSP004H genomes. The inner light red ring represents the NSP009P genome; the middle red one shows the SP165 genome after BLASTn match; while the outer green, light blue and dark blue rings correspond to NSP023P, NSP003P and NSP004H, respectively. Missing regions identified by the BLAST analysis on the CGView server’s PROKSEE software are shown as ‘gaps’ on each of the circular genomes. Arrows represent regions shared between SP165 and NSP009P but not present in the NSP023P, NSP003P and NSP004H genomes. The inner ring corresponds to the GC skew of *S. pettenkoferi* NSP009P isolate. Light grey profile indicates overabundance of GC nucleotides, whereas dark grey shows the opposite. The inner black ring represents the GC%.

**Table 1 ijms-23-15476-t001:** Characteristics and resistance profiles of the studied *S. pettenkoferi* strains (*n* = 29).

Isolate ID	Year	Type of Sampling	Hospital Department	Patients		Resistance Profile
				Age	Sex	
SP165	2018	Tissue biopsy	Diabetic foot clinic	60	M	PEN, OXA, ERY, CMN, L, OFX, RIF, FOS
NSP003P	2020	Tissue biopsy	Diabetic foot clinic	58	M	/
NSP004P	2020	Tissue biopsy	Diabetic foot clinic	89	M	OFX, FA, FOS
NSP007P	2020	Bone biopsy	Diabetic foot clinic	69	M	PEN, FA
NSP008P	2020	Bone biopsy	Vascular surgery	90	M	FOS
NSP009P	2020	Bone biopsy	Diabetic foot clinic	62	M	PEN, ERY, CMN, L, PT, SYN, TET, OFX, FA, FOS
NSP010P	2020	Bone biopsy	Orthopaedic surgery	61	F	FOS
NSP011P	2020	Tissue biopsy	Diabetic foot clinic	89	M	/
NSP012P	2021	Bone biopsy	Orthopaedic surgery	67	M	PEN, OFX, FA
NSP013P	2021	Tissue biopsy	Diabetic foot clinic	94	M	PEN, OFX
NSP015P	2021	Bone biopsy	Orthopaedic surgery	64	M	FOS
NSP016P	2021	Bone biopsy	Diabetic foot clinic	56	M	FA
NSP017P	2021	Tissue biopsy	Diabetic foot clinic	84	M	/
NSP018P	2021	Bone biopsy	Vascular surgery	76	M	/
NSP019P	2021	Bone biopsy	Vascular surgery	71	M	FOS
NSP020P	2021	Tissue biopsy	Orthopaedic surgery	63	M	/
NSP021P	2021	Bone biopsy	Diabetic foot clinic	60	M	PEN, FOS
NSP022P	2021	Tissue biopsy	Diabetic foot clinic	66	F	FA
NSP023P	2021	Bone biopsy	Orthopaedic surgery	94	F	PEN
NSP024P	2021	Tissue biopsy	Diabetic foot clinic	78	F	PEN
NSP025P	2022	Tissue biopsy	Diabetic foot clinic	93	M	/
NSP026P	2022	Tissue biopsy	Diabetic foot clinic	63	M	PEN
NSP001H	2020	Blood culture	Geriatric	83	M	PEN, OFX, FA, FOS
NSP002H	2020	Blood culture	Emergency	59	F	ERY, FOS
NSP003H	2020	Blood culture	Emergency	65	M	FA, FOS
NSP004H	2021	Blood culture	Polyvalent medicine	44	F	PEN
NSP005H	2021	Blood culture	Intensive care unit	58	M	FOS
NSP007H	2022	Blood culture	Neurology	72	M	PEN, FOS
NSP008H	2022	Blood culture	Intensive care unit	62	M	FOS

F, female; M, male; PEN, penicillin G; OXA, oxacillin; ERY, erythromycin; CMN, clindamycin; L, lincomycin; PT, pristinamycin; SYN, synergystin; TET, tetracyclin; OFX, ofloxacin; FA, fusidic acid; RIF, rifampicin; FOS, fosfomycin.

**Table 2 ijms-23-15476-t002:** Comparison of resistomes between blood culture and DFU *S. pettenkoferi* strains.

Resistance Genes	Blood Culture Strains (*n* = 7)	DFU Strains (*n* = 22)	Total (*n* = 29)	*p*-Value
				Blood vs. DFU
*blaZ*	3 (42.9%)	9 (40.9%)	12 (41.4%)	1
*mecA*	0 (0%)	1 (4.5%)	1 (3.4%)	1
*ermA*	0 (0%)	3 (13.6%)	3 (10.3%)	0.56
*vgaA*	3 (42.9%)	3 (13.6%)	6 (20.7%)	0.13
*gyrA*	1 (14.3%)	5 (22.7%)	6 (20.7%)	1
*grlA*	6 (87.5%)	17 (77.2%)	23 (79.3%)	1
*grlB*	1 (14.3%)	13 (59%)	14 (48.3%)	0.08
*tetk*	0 (0%)	1 (4.5%)	1 (3.4%)	1
*rpoB*	0 (0%)	1 (4.5%)	1 (3.4%)	1
*fusB*	1 (14.3%)	7 (31.8%)	8 (27.6%)	0.63
*ant(9)-Ia*	0 (0%)	3 (13.6%)	3 (10.3%)	0.56
*fosB*	4 (57.1%)	6 (27.3%)	11 (37.9%)	0.19

Fisher’s exact test was used to establish statistical significance.

**Table 3 ijms-23-15476-t003:** Comparison of biofilm-related genes between the *S. pettenkoferi* blood culture and DFU strains groups and biofilm groups.

Biofilm Steps		Putative Genetic Determinants of Virulence		Blood Culture (*n* = 7)	DFU (*n* = 22)	*p*-Value B vs. D	Biofilm Group 1 (*n* = 17)	Biofilm Group 2 (*n* = 9)	Biofilm Group 3 (*n* = 3)	*p*-Value G1 vs. G2 vs. G3	Total (*n* = 29)
**INITIAL ADHESION**	To abiotic surfaces	*bhp*	Biofilm associated protein	0 (0%)	2 (9%)	1	0 (0%)	1 (11%)	1 (33%)	0.07	2 (6.9%)
		*atlE*	Autolysin E	7 (100%)	22 (100%)	-	17 (100%)	9 (100%)	3 (100%)	-	29 (100%)
		*sesC, sasC, sasG*	Surface proteins	0 (0%)	0 (0%)	-	0 (0%)	0 (0%)	0 (0%)	-	0 (0%)
		*embp*	Extracellular matrix binding protein Embp	0 (0%)	0 (0%)	-	0 (0%)	0 (0%)	0 (0%)	-	0 (0%)
	To biotic surfaces	*fnbpA, fnbpB*	Fibronectin binding protein A and B	0 (0%)	0 (0%)	-	0 (0%)	0 (0%)	0 (0%)	-	0 (0%)
		*fbl, fib*	Fibrinogen binding protein	0 (0%)	0 (0%)	-	0 (0%)	0 (0%)	0 (0%)	-	0 (0%)
		*clfA, clfB*	Clumping factors A and B	0 (0%)	0 (0%)	-	0 (0%)	0 (0%)	0 (0%)	-	0 (0%)
		*sdrCDEFH*	Serine-aspartate repeat-containing protein C,D,E,F,H	0 (0%)	0 (0%)	-	0 (0%)	0 (0%)	0 (0%)	-	0 (0%)
		*sdrD B-like domain*	B-like domain from the SdrD protein	4 (57.1%)	5 (22.7%)	0.15	7 (41.2%)	1 (11%)	1 (33%)	0.24	9 (31%)
**MATURATION**	Intercellular aggregation	*icaA, icaB, icaC*	Polysaccharide Intercellular Adhesion (PIA)	7 (100%)	22 (100%)	-	17 (100%)	9 (100%)	3 (100%)	-	29 (100%)
		*icaD*	Polysaccharide Intercellular Adhesion (PIA)	0 (0%)	0 (0%)	-	0 (0%)	0 (0%)	0 (0%)	-	0 (0%)
		*isdA, isdB*	Iron-regulated surface determinant protein A,B	0 (0%)	0 (0%)	-	0 (0%)	0 (0%)	0 (0%)	-	0 (0%)
		*aap*	Accumulation associated protein	0 (0%)	0 (0%)	-	0 (0%)	0 (0%)	0 (0%)	-	0 (0%)
		*sbp*	*S. epidermidis* biofilm accumulation	0 (0%)	0 (0%)	-	0 (0%)	0 (0%)	0 (0%)	-	0 (0%)
	Biofilm structuration	*splABCDE*	Serine proteases SplA, SplB, SplC, SplD, SplE	0 (0%)	0 (0%)	-	0 (0%)	0 (0%)	0 (0%)	-	0 (0%)
		*splF*	Serine protease SplF	7 (100%)	22 (100%)	-	17 (100%)	9 (100%)	3 (100%)	-	29 (100%)
		*clpP*	ClpP protease	7 (100%)	22 (100%)	-	17 (100%)	9 (100%)	3 (100%)	-	29 (100%)
		*SSP1763*	Serine protease HtrA-like	7 (100%)	22 (100%)	-	17 (100%)	9 (100%)	3 (100%)	-	29 (100%)
		*HMPREF2802_03390*	Serine protease	7 (100%)	22 (100%)	-	17 (100%)	9 (100%)	3 (100%)	-	29 (100%)
**REGULATION**		*icaR*	Biofilm operon *icaADBC* regulator	0 (0%)	0 (0%)	-	0 (0%)	0 (0%)	0 (0%)	-	0 (0%)
		*agrA, agrC*	Accessory genes regulators A and C	7 (100%)	22 (100%)	-	17 (100%)	9 (100%)	3 (100%)	-	29 (100%)
		*agrB, agrD*	Accessory gene sregulators B and D	0 (0%)	0 (0%)	-	0 (0%)	0 (0%)	0 (0%)	-	0 (0%)
		*sigB*	Sigma B	7 (100%)	22 (100%)	-	17 (100%)	9 (100%)	3 (100%)	-	29 (100%)
		*rsbUVW*	/	7 (100%)	22 (100%)	-	17 (100%)	9 (100%)	3 (100%)	-	29 (100%)
		*lytSR*	LytSR two component regulatory system	7 (100%)	22 (100%)	-	17 (100%)	9 (100%)	3 (100%)	-	29 (100%)
		*arlRS*	ArlRS two-component system	7 (100%)	22 (100%)	-	17 (100%)	9 (100%)	3 (100%)	-	29 (100%)
		*mgrA*	HTH-type transcription regulator MgrA	7 (100%)	22 (100%)	-	17 (100%)	9 (100%)	3 (100%)	-	29 (100%)
		*cidA*	Holin-like protein CidA	6 (85.7%)	17 (77.3%)	1	12 (70.6%)	9 (100%)	2 (67%)	0.22	23 (79.3%)
		*lrgA*	Antiholin-like protein LrgA	7 (100%)	22 (100%)	-	17 (100%)	9 (100%)	3 (100%)	-	29 (100%)
		*sarA*	Transcriptional regulator SarA	7 (100%)	22 (100%)	-	17 (100%)	9 (100%)	3 (100%)	-	29 (100%)
		*luxS*	S-ribosylhomocysteine lyase	7 (100%)	22 (100%)	-	17 (100%)	9 (100%)	3 (100%)	-	29 (100%)
**DISPERSION**		*BUY34_08050*	Hemolysin H3C: Phenol soluble modulin beta protein	7 (100%)	22 (100%)	-	17 (100%)	9 (100%)	3 (100%)	-	29 (100%)

Fisher’s exact test was used to establish statistical significance.

**Table 4 ijms-23-15476-t004:** Comparison of virulence-related genes between *S. pettenkoferi* isolated from blood cultures and infected DFUs and their biofilm group profiles.

	Putative Genetic Determinants of Virulence		Blood Culture (*n* = 7)	DFU (*n* = 22)	*p*-Value B vs. D	Biofilm Group 1 (*n* = 17)	Biofilm Group 2 (*n* = 9)	Biofilm Group 3 (*n* = 3)	*p*-Value G1 vs. G2 vs. G3	Total (*n* = 29)
**Toxins**	*yqfA*	Hemolysin III	7 (100%)	22 (100%)	-	17 (100%)	9 (100%)	3 (100%)	-	29 (100%)
	*hld*	Delta hemolysin	5 (71.4%)	18 (81.8%)	0.61	14 (82.4%)	6 (66.7%)	3 (100%)	0.53	23 (79.3%)
	*ERS010499_02103*	Hemolysin H1U	6 (85.7%)	9 (40.9%)	0.0801	11 (65.7%)	4 (44.4%)	0 (0%)	0.1218	15 (51.7%)
	*hly (hla), hlb, hlg*	α- β- γ-hemolysins	0 (0%)	0 (0%)	-	0 (0%)	0 (0%)	0 (0%)	-	0 (0%)
	*sea, sel, seg, sed*	Enterotoxins	0 (0%)	0 (0%)	-	0 (0%)	0 (0%)	0 (0%)	-	0 (0%)
	*tst-1*	Toxic shock syndrome toxin	0 (0%)	0 (0%)	-	0 (0%)	0 (0%)	0 (0%)	-	0 (0%)
	*etA, etB, etD*	Exfoliatin	0 (0%)	0 (0%)	-	0 (0%)	0 (0%)	0 (0%)	-	0 (0%)
	*lukDE*	Leukotoxin LukD and LukE	0 (0%)	0 (0%)	-	0 (0%)	0 (0%)	0 (0%)	-	0 (0%)
**Avoid Host Immune Response and Invasiveness**	*capA*	Capsule biosynthesis protein CapA	7 (100%)	22 (100%)	-	17 (100%)	9 (100%)	3 (100%)	-	29 (100%)
	*eno*	Enolase	7 (100%)	22 (100%)	-	17 (100%)	9 (100%)	3 (100%)	-	29 (100%)
	*katA*	Catalase	7 (100%)	22 (100%)	-	17 (100%)	9 (100%)	3 (100%)	-	29 (100%)
	*/*	esXA: Type VII secretion system (T7SS) substrate	2 (28.6%)	4 (18.2%)	0.61	4 (23.5%)	1 (11.1%)	1 (33%)	0.67	6 (20.7%)
	*T7SSB apparatus encoding genes*	T7SSb apparatus: EsaA, EsaB, EssA, EssB and EssC	1 (14.3%)	1 (4.5%)	0.43	1 (5.9%)	1 (11.1%)	0 (0%)	1	2 (6.9%)
	*tuf*	Elongation factor tu	7 (100%)	22 (100%)	-	17 (100%)	9 (100%)	3 (100%)	-	29 (100%)
	*lspA*	Lipoprotein signal peptidase	7 (100%)	22 (100%)	-	17 (100%)	9 (100%)	3 (100%)	-	29 (100%)
	*lgt*	Lipoprotein diacylglycerol transferase	7 (100%)	22 (100%)	-	17 (100%)	9 (100%)	3 (100%)	-	29 (100%)
	*nucH*	Thermonuclease H	7 (100%)	22 (100%)	-	17 (100%)	9 (100%)	3 (100%)	-	29 (100%)
	*coA*	Coagulase	0 (0%)	0 (0%)	-	0 (0%)	0 (0%)	0 (0%)	-	0 (0%)
	*spA*	*Staphylococcus* protein A	0 (0%)	0 (0%)	-	0 (0%)	0 (0%)	0 (0%)	-	0 (0%)
	*vwb*	Von Willebrand factor binding protein Vwb	0 (0%)	0 (0%)	-	0 (0%)	0 (0%)	0 (0%)	-	0 (0%)
	*saK*	Staphylokinase	0 (0%)	0 (0%)	-	0 (0%)	0 (0%)	0 (0%)	-	0 (0%)
**Protease activity or implicated in proteolysis**	*clpC, clpX, clpXP*	Clp protases	7 (100%)	22 (100%)	-	17 (100%)	9 (100%)	3 (100%)	-	29 (100%)
	*mecA*	Adapter protein MecA	7 (100%)	22 (100%)	-	17 (100%)	9 (100%)	3 (100%)	-	29 (100%)
	*aur*	Aureolysin	0 (0%)	0 (0%)	-	0 (0%)	0 (0%)	0 (0%)	-	0 (0%)
**Regulation**	*nsaS*	Sensor histidine kinase	7 (100%)	22 (100%)	-	17 (100%)	9 (100%)	3 (100%)	-	29 (100%)
	*nsaR*	DNA-binding response regulator	0 (0%)	0 (0%)	-	0 (0%)	0 (0%)	0 (0%)	-	0 (0%)
	*srrB*	Sensor protein SrrB	7 (100%)	22 (100%)	-	17 (100%)	9 (100%)	3 (100%)	-	29 (100%)
	*srrA*	Transcriptional regulatory protein SrrA	0 (0%)	0 (0%)	-	0 (0%)	0 (0%)	0 (0%)	-	0 (0%)
	*vraSR*	Sensor Pt VraS/ Response regulator pt VraR	7 (100%)	22 (100%)	-	17 (100%)	9 (100%)	3 (100%)	-	29 (100%)
	*saeSR*	Histidine protein kinase SaeS/ Response regulator SaeR	7 (100%)	22 (100%)	-	17 (100%)	9 (100%)	3 (100%)	-	29 (100%)
**Others**	*virE*	Virulence-associated protein E	0 (0%)	2 (9.1%)	1	0 (0%)	1 (11%)	1 (33%)	0.07	2 (6.9%)
	*cvfB, cvfC*	Conserved virulence factor B and C	7 (100%)	22 (100%)	-	17 (100%)	9 (100%)	3 (100%)	-	29 (100%)
	*CJ235_09000*	α- β- hydrolase	7 (100%)	22 (100%)	-	17 (100%)	9 (100%)	3 (100%)	-	29 (100%)
	*isdF*	Iron-regulated surface determinant protein F	7 (100%)	22 (100%)	-	17 (100%)	9 (100%)	3 (100%)	-	29 (100%)

## Data Availability

Data supporting reported results can be found in the file “S. pettenkoferi project” contained in the computer of the VBIC unit.

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
