# Peer review of "Phenotypic and Genotypic Virulence Characterisation of *Staphylococcus pettenkoferi* Strains Isolated from Human Bloodstream and Diabetic Foot Infections"

_ijms, 2022, doi:10.3390/ijms232415476_

Round 1

Reviewer 1 Report

This study characterized twenty-nine clinical isolates of Staphylococcus pettenkoferi that are important in diabetic foot ulcer (DFU) infection. The study explored the genome of the bacteria along with the phenotypic characterization. I have a few suggestions as follows.

1.                   Please confirm which strain [CIP 107711 (text) or CCUG51270 (figure1)] was used for analysis.

2.                   Please confirm the number of other clinical strains isolated from DFU in subclade I-b.

3.                   This study explored whole genome sequencing which the authors can analyze the data more than the presence or absence of the target genes. SNP analysis can be done. The association between the genome and the phenotype of the bacteria can be reported in detail.

4.                   This study compared the ability of biofilm formation among clinical isolates using a non-biofilm-producing strain (CIP107711) as a control that whole genome sequencing data is available. The comparative genome among these groups can be done

Author Response

Reviewer 1:

  1. Please confirm which strain [CIP 107711 (text) or CCUG51270 (figure1)] was used for analysis.

We confirm that the two denominations correspond to only on strain. We modified the Figure 1 in the new version of the manuscript.

  1. Please confirm the number of other clinical strains isolated from DFU in subclade I-b.

We modified the sentence and included the 15 strains belonging to subclade I-b in the main text (Ln 80-83).

  1. This study explored whole genome sequencing which the authors can analyze the data more than the presence or absence of the target genes. SNP analysis can be done. The association between the genome and the phenotype of the bacteria can be reported in detail. 

To date, the genomes of S. pettenkoferi have been scarcely described. In this paper, we focused on the detection of the main genes involved in the virulence and resistance of Staphylococcus sp. Following the reviewer request, we added the SNP analysis of the main studied strains (NSP003P, NSP009P, NSP023P, NSP004H and SP165). This analysis confirmed the genomic diversity between these five strains (Figures S1 and S2). A pangenomic analysis was also added, showing that the two most virulent strains (NSP009P and SP165) have clustered together. The association between genome and phenotype has been described (with the genes involved in biofilm and virulence) (Tables S5 and S6). Conserved coding regions between all strains required further investigation.

  1. This study compared the ability of biofilm formation among clinical isolates using a non-biofilm-producing strain (CIP107711) as a control that whole genome sequencing data is available. The comparative genome among these groups can be done. 

 The analysis of genes involved in biofilm and virulence has been described. We added the same analysis concerning CIP107711 strain. The supplementary data (Tables S6 and S7) present the distribution of biofilm and virulence genes between the different strains and CIP107711.

Reviewer 2 Report

The manuscript presents thorough and intriguing analysis of Staphylococcus pettenkoferi strains and their virulence. Some minor details could be improved:

Fig.1: CIP 107711 is probably CCUG51270 in the dendrogram, yet this is not clarified.

In 2.3.1. the strains were assigned to three groups on their ability to form biofilm. I was not able to find the exact group for each strain. It would be appropriate to include this.

Fig.2 "Dotted horizontal lines" seem to be missing from the graph.

In 2.6, the text states the more virulent strain NSP009P didn't differ from less virulent strains in virulome content, yet the reader can not easily compare them, except of very general genome representation in Fig.5. It would be appropriate to show the genes from Table 4 as distributed among the strains.

Methods, 4.2: "All the identified genomics sequences have been deposited on GenBank website accession bioproject: PRJNA870410". Will it be released on publication date?

Author Response

Reviewer 2

- Fig.1: CIP 107711 is probably CCUG51270 in the dendrogram, yet this is not clarified.

A new Figure 1 has been added in the revised version of our manuscript.

- In 2.3.1. the strains were assigned to three groups on their ability to form biofilm. I was not able to find the exact group for each strain. It would be appropriate to include this.

We modified accordingly.

- Fig.2 "Dotted horizontal lines" seem to be missing from the graph.

We modified the Figure 2 and added the 3 biofilm groups (according to the previous comment).

- In 2.6, the text states the more virulent strain NSP009P didn't differ from less virulent strains in virulome content, yet the reader can not easily compare them, except of very general genome representation in Fig.5. It would be appropriate to show the genes from Table 4 as distributed among the strains.

We added information in Supplementary data Table S5 and S6 with the distribution of biofilm and virulence genes between the different strains and CIP 107711.

- Methods, 4.2: "All the identified genomics sequences have been deposited on GenBank website accession bioproject: PRJNA870410". Will it be released on publication date?

We confirm that the genomic sequences will be released on publication date.